# ARB: Advanced Reasoning Benchmark for Large Language Models

## Abstract

Large Language Models (LLMs) have demonstrated remarkable performance on various quantitative reasoning and knowledge benchmarks. However, many of these benchmarks are losing utility as LLMs get increasingly high scores, despite not yet reaching expert performance in these domains. We introduce ARB, a novel benchmark composed of advanced reasoning problems in multiple fields. ARB features problems in mathematics, physics, biology, chemistry, and law. As a subset of ARB, we introduce a challenging set of math and physics problems which require advanced symbolic reasoning and domain knowledge. We evaluate recent models such as GPT-4 and Claude on ARB and demonstrate that current models score well below 50% on more demanding tasks. In order to improve both automatic and assisted evaluation capabilities, we introduce a rubric-based evaluation approach, allowing GPT-4 to score its own intermediate reasoning steps. Further, we conduct a human evaluation of the symbolic subset of ARB, finding promising agreement between annotators and GPT-4 rubric evaluation scores.

## 1 Introduction

In recent years, models such as GPT-3 (Brown et al., 2020), GPT-4 (OpenAI, 2023), PaLM (Chowdhery et al., 2022), and Chinchilla (Hoffmann et al., 2022) have shown increasing performance across a wide variety of natural language tasks ranging from translation to reasoning (Bubeck et al., 2023; Laskar et al., 2023). This rapid progress has been closely tracked and assessed by evaluating LLMs on benchmarks, which test model capabilities on a set of standardized problems. The GLUE benchmark (Wang et al., 2019b) for language understanding was first released in April 2018; but models such as BERT (Devlin et al., 2019) and GPT-2 (Radford et al., 2019) in the following year were already powerful enough to necessitate the "SuperGLUE" benchmark (Wang et al., 2019a). Since then, the race between language models and benchmarks has increasingly favored the former.

Scaling up, model sizes and datasets alike, has led to rapid improvements on various natural language tasks on benchmarks like BIG-bench (Srivastava et al., 2022) and HELM (Liang et al., 2022). Neural scaling laws (Kaplan et al., 2020; Caballero et al., 2023; Alabdulmohsin et al., 2022) were used to predict the behavior of large scale models on various metrics. Nevertheless, LLM performance often increases unpredictably (Wei et al., 2022a), especially on tasks that require reasoning abilities. Predictions of performance on ML benchmarks often underestimate the rate of progress (Steinhardt, 2022). Since progress has been faster than anticipated, new benchmarks need to be more difficult.

Models such as ChatGPT have shown the ability to pass entry-level examinations in fields such as law (Bommarito II and Katz, 2022), medicine (Kung et al., 2023), economics (Caplan, 2023), and mathematics (Shakarian et al., 2023). Nevertheless, LLM understanding of many fields is reportedly shallow and unreliable (Shapira et al., 2023). *Expert reasoning* in domains with specialized knowledge is essential for automated systems to augment skilled professionals (Noy and Zhang, 2023).

In this paper, we introduce a new benchmark dataset, **ARB** (**A**dvanced **R**easoning **B**enchmark), designed to evaluate expert reasoning abilities in mathematics, physics, chemistry, biology, and law. To make the benchmark more challenging than previous benchmarks, we extract graduate-level tasks from resources intended for domain professionals. The mathematics and physics portions are more difficult than popular benchmarks such as MATH (Hendrycks et al., 2021), due to both the content and the question format. The performance of current models such as GPT-4 on the quantitative parts of ARB is very low using standard prompting methods.

Our dataset offers improvements over existing benchmarks:

- Hundreds of problems requiring expert reasoning in quantitative subjects, where LLMs are known to underperform;
- For mathematics and physics, all problems are short-answer and open-response questions, in contrast to the multiple-choice questions that dominated earlier benchmarks.

In addition, we propose an automated rubric-based method allowing self-evaluation of intermediate reasoning steps. While not currently a substitute for human evaluation, rubrics generated by GPT-4 have good coverage, and self-evaluation scores track human grading surprisingly well.

We provide the instructions to access the dataset in the supplementary material.

## 2 RELATED WORK

Improving the reasoning capabilities of LLMs had been a subject of recent interest, with a particular focus on advanced prompting techniques (Wei et al., 2022b; Kojima et al., 2023; Wang et al., 2023; Yao et al., 2023; Nye et al., 2021). Such techniques have seen increasingly successful applications in solving reasoning problems involving commonsense reasoning and mathematics, by promoting active reasoning processes within the LLMs before yielding final answers.

Model architectures such as Minerva (Lewkowycz et al., 2022) have exemplified the enhancement of reasoning capabilities through fine-tuning on extensive datasets covering math and reasoning tasks. This has yielded improved performance across several benchmarks, including MATH (Hendrycks et al., 2021), GSM8K (Cobbe et al., 2021), and MMLU (Hendrycks et al., 2020). Concurrently, other lines of research (Li et al., 2023; Lightman et al., 2023; Cobbe et al., 2021) have investigated the application of verification techniques to augment and enhance LLM performance.

Most of the aforementioned work has typically evaluated techniques against math benchmarks (e.g., GSM8K (Cobbe et al., 2021), MATH (Hendrycks et al., 2021), SVAMP (Patel et al., 2021), ASDiv (Miao et al., 2020), AQuA (Ling et al., 2017), MAWPS (Koncel-Kedziorski et al., 2016), MultiArith (Roy and Roth, 2016)) and commonsense reasoning tasks (e.g., CSQA (Talmor et al., 2018), StrategyQA (Geva et al., 2021), HotpotQA (Yang et al., 2018)). Recently, several new benchmarks have been introduced for reasoning and planning tasks, such as the GPT-Planning Benchmark (Valmeekam et al., 2023), ALERT Reasoning Benchmark (Yu et al., 2022), and (Gendron et al., 2023). Additionally, comprehensive evaluation suites like the Chain-of-Thought Hub (Fu et al., 2023) have been proposed. Particularly related to our work is JEEBench (Arora et al., 2023)), which tests some of the same models as we do, on mathematics, physics and chemistry tasks. The main differences with our work are our quantitative problems are both somewhat harder and require deeper math/physics knowledge, and that their benchmark is entirely multiple-choice.

Most existing benchmarks are limited in difficulty and represent a restricted range of reasoning tasks. Moreover, recent advancements such as Minerva (Lewkowycz et al., 2022) have revealed that these benchmarks may not offer sufficient challenge. Of course, no single paper can solve these issues by itself; evaluation is co-evolving with capabilities and new benchmarks are always needed.

The rapid progress in LLM capabilities has led many to explore using LLMs in the LLM evaluation pipeline. Apart from using LLMs to generate evaluation tasks (Zhang et al., 2022; Perez et al., 2022), LLMs have increasingly been used as a proxy for human evaluation (Chiang and Lee, 2023; Liu et al., 2023; Fu et al., 2023; Kocmi and Federmann, 2023). Useful LLM-based evaluation for alignment has been done using rubrics (Bai et al., 2022). We explore the efficacy of rubrics for evaluation when applied to highly complex math and physics problems, proposing a path forward to issues discussed in Arora et al. (2023).

## 3 BENCHMARK

The key considerations when building an LLM benchmark are:

- **Difficulty.** Most tasks have to be out of reach of current models; a benchmark where many models score over 90% is not useful for tracking differential AI development.

- **Usefulness.** The tested skills should correlate with generally useful human skills.
- **Ease of evaluation.** It should be straightforward for the model creators to compare the performances of different models. The scores should be interpretable.
- **Minimizing data contamination.** A consistent issue with popular benchmarks is that recent LLMs contain some tasks in their training data (OpenAI, 2023). This leads to overestimation of true model capabilities.
- **Connection to general capabilities.** If a model is trained on data similar to the benchmark, it is possible it achieves high performance without generalization or "intelligence", failing to solve novel tasks of similar difficulty (Chollet, 2019). Conversely, problems should not be pathological or overly adversarial, to avoid the dangers of underclaiming (Bowman, 2021).

The main component of this benchmark is the quantitative portion, i.e. the math and physics problems; those help provide a test suite that is difficult enough to differentiate between the capabilities of state-of-the-art LLMs. The law and MCAT portions of the dataset are complementary, helping assess the model's capabilities beyond quantitative tasks, on areas that are popular and important application domains for LLMs.

## 3.1 Output Formats

The benchmark consists of three types of questions: multiple choice, short answer, and open response, in descending order of proportion in the dataset.

- **Multiple choice** questions consist of a question and four to five possible answers, and the correct answer is the one that best answers the question. Those were sourced from standardized tests, such as the MCAT and bar exam prep, and make up a large proportion of the dataset due to their ease of grading.
- **Short answer questions**, on the other hand, ask for final answers in the format of a short phrase or mathematical expression. They were sourced from problem books such as Souza and Silva (2008), Gelca and Andreescu (2017), and physics book series Lim and Qiang (2001), Lim (2007), Lim (1998), Lim et al. (2019), and Lim (1996). We generally avoided algebraic expressions, because of technical difficulties in the grading process.

  A given algebraic expression may have several equivalent forms (e.g., nontrivial functional relations for the functions appearing in the final answer), and a grading scheme which accounts for all possible variations across our entire dataset is not feasible. Moreover, physics problems often require answers introducing new notation that is not explicitly mentioned in the problem statement.
- **Open response** questions are more challenging: those consist of a question and a blank space for the answer. Those were sourced from problem books and exams, such as the Harvard PhD comprehensive exams in mathematics (Harvard University, 2021). Such tasks require manual grading. On these, GPT-4 rarely produces satisfactory responses, even when only elementary knowledge is required.

## 3.2 Mathematics

This part of the dataset is the most diverse. It includes contest mathematics problems as well as "university mathematics" (i.e. mathematics traditionally taught in universities at the undergraduate and beginning graduate level). Contest problems are sourced from Gelca and Andreescu (2017) and Brayman and Kukush (2018), and university mathematics problems are sourced from Souza and Silva (2008), Chen and Li (1998) and Harvard University (2021). The dataset does not include high school contest problems because those are already covered in other well-known benchmarks (Hendrycks et al., 2021). The Putnam and Brayman books both contain official solutions, which we also include in the dataset. This can be useful for automating the grading process, which we explore in Section 5.

For university mathematics, we pick Souza and Silva (2008) and Chen and Li (1998) for its large selection of "standard" undergraduate mathematics problems, as well as many problems suitable for the short answer portions. We also select Harvard University (2021) because it covers topics that other collections of exams rarely cover, such as representation theory of finite groups and algebraic topology.

Table 1: Types of problems in the benchmark by subject area.

| Subject | Answer Type | Number |
|---|---|---|
| Physics | Numerical | 113 |
| | Numerical (w/ image) | 18 |
| | Symbolic | 51 |
| | Symbolic (w/ image) | 13 |
| Mathematics | Numerical | 69 |
| | Symbolic | 52 |
| | Proof-like | 19 |
| Law | Multiple Choice | 627 |
| MCAT (Reading) | Multiple Choice | 165 |
| MCAT (Science) | Multiple Choice | 144 |
| | Multiple Choice (w/ image) | 37 |

The mathematics problems on our benchmark are significantly harder than existing benchmarks because of both the mathematical content and the way our problems are posed. To take some popular examples, the MATH dataset consists of pre-olympiad high school competition problems (AMC 10, AMC 12, and AIME) which only use pre-calculus techiques and always have numerical final answers. The hardest problems on the MMLU dataset are in the College Mathematics and Abstract Algebra sections, which are at the level of the GRE exams (the general and math subject portions, respectively). The BIG-Bench dataset contains several mathematical tasks, including *chinese_remainder_theorem* and *mathematical_induction*, most of which require at most high school mathematics. The most advanced task in the benchmark is likely *identify_math_theorems*, because it requires understanding of some advanced mathematical terms; but all problems can be solved by a process of elimination, which cannot work on our benchmark.

### 3.3 PHYSICS

The physics problems are structured similarly as the math problems. The main difference is that some physics problems contain figures, and there are more problems with numerical answers. The problems were sourced from the Major American Universities PhD Qualifying Questions and Solutions series (Zhongguo-Kexue-Jishu-Daxue, 1990).

### 3.4 MCAT

The MCAT test contains multiple choice problems testing biology, psychology, chemistry, physics, and reading comprehension. The MCAT problems are sampled from the third edition of McGraw-Hill Education 3 MCAT Practice Tests (Campbell et al., 2017) and cover both science and reading questions. This book was chosen as very few of these problems appear in standard web-searchable sources, limiting contamination. As in the previous categories, we pick problems which are self-contained. Because some MCAT science questions are accompanied by images, we accompany such questions with corresponding image files.

### 3.5 LAW

Application of legal knowledge to a particular scenario requires logical reasoning. This makes assessments of legal skills an especially attractive type of language model benchmark, where we are attempting to assess the reasoning and intelligence of these models. Furthermore, if the models better understand law, they can be more reliable and ultimately more useful in real-world applications, potentially even increasing the efficiency and transparency of governments more broadly.

Most lawyers in the U.S. go to law school, graduate, then study for the Bar Examination, and then must pass the bar before going on to practice law professionally. To evaluate legal understanding of the models, we use an older Bar Examination practice set that is less likely to be available online in a way that could have led to its legal inclusion in training data for the language models that we are

assessing. The practice bar exam we administer to the language models covers most major areas of law, and tests legal reasoning and broad U.S. legal knowledge.

## 4 EVALUATION

We evaluated current LLMs on all text-only problems in our dataset. Other LLM benchmark papers do not evaluate on multimodal tasks due to the lack of good multimodal models; we follow suit. Given public communications about GPT-4 (OpenAI, 2023) and Gemini (Ghahramani, 2023), it is likely the physics and MCAT image problems will be useful for testing multimodal LLMs soon.

**Models**   We evaluate ChatGPT (`gpt3.5-turbo-0301`), GPT 3.5 (`text-davinci-003`), GPT-4 with 8k context length (`gpt-4-0314`), and Claude (`claude-v1.3-100k`). We use task-specific instructions and chain of thought for all question types. In chat models, we placed the instructions as the system prompt; otherwise, we put them at the beginning of the prompt. Temperature was set to 0.7, unless noted otherwise.

In all problem types, in order to extract the model's final answer, we instruct the model to write its final answer at the end of the response after the delimiter `ANSWER:` . We then parse the model generated final answer as the remaining text after the delimiter. The response is marked as incorrect if the delimiter is not found. Due to the differences in evaluation for multiple choice versus open-ended responses, we adopt several evaluation procedures.

**Multiple choice**   To evaluate multiple choice questions, we can simply compare the extracted final answer to the ground truth. A response is considered correct if the extracted choice matches the ground truth choice. We conducted a separate manual evaluation on a sampled subset of the questions to check that our parsing procedure is not mischaracterizing the true performance of the model.

**Numerical**   To evaluate problems with a numerical final answer, we first extracted the delimited model answer as above. In the physics problems, many answers are in units; we prompt the model with information about the unit, and instruct it to fully simplify its answer and omit any units. However, sometimes the model forgets to do either or both, and so we apply a series of regexes to remove units. We then attempt to parse the result into a mathematical expression using Python's SymPy library (Meurer et al., 2017). If this parsing fails, the answer is marked as incorrect. Once parsed, we scored a the model answer as correct if $\frac{|\text{model\_answer}-\text{ground\_truth}|}{\text{ground\_truth}} < 0.01$.

**Symbolic**   Problems with symbolic answers are less structured and harder to parse. To do so, we again leverage SymPy, first normalizing expressions to contain a default set of variable names and then checking for equivalence up to a permutation of the variables. However this approach is error-prone and only works for the subset of symbolic responses in a function form. More advanced responses, such as those containing set notation, require human evaluation.

**Proof-like**   Natural language proofs cannot be evaluated automatically; the authors with training in mathematics grade the proofs. Further manual human evaluation requires a thorough inspection of the intermediate reasoning steps. This makes evaluation expensive in practice.

**Model-based evaluation**   To address the difficulties in developing automated metrics for evaluating more advanced problems, we experiment with two model based approaches. First, we prompt ChatGPT to grade the equivalence of two symbolic expressions with score options $0$ when the totally incorrect, $0.5$ when the symbolic expressions are nearly the same e.g. equivalent up to a constant, and $1$ when they are an exact match. Our prompting strategy can be found in the supplementary material.

More generally, we evaluate the capabilities of GPT-4 to grade intermediate reasoning chains via a *rubric-based* evaluation approach. For symbolic and proof-like problems, we few-shot prompt GPT-4 to create a 10-point rubric. This is done by handwriting a small set of initial rubrics for proof-like problems and prompting the model with these examples and the ground truth reference solution. The model assigns point values to intermediate steps using the reference solution as a guide. This process is illustrated in the supplementary material.

With model generated rubrics in hand, we then evaluate each question against its rubric. This is done by again prompting GPT-4 to go step by step through the model answer and assign partial credit based on the rubric. This provides a denser automatic evaluation metric on increasingly unstructured answers. As a nice side benefit, it makes human evaluation of complex symbolic questions much easier, significantly reducing the amount of time required per question.

## 4.1 RESULTS

We now discuss the evaluation of `gpt-4`, `gpt-3.5-turbo`, `text-davinci-003`, and `claude-v1.3` on ARB. The results for the mechanically scored subjects are in Figure 1.

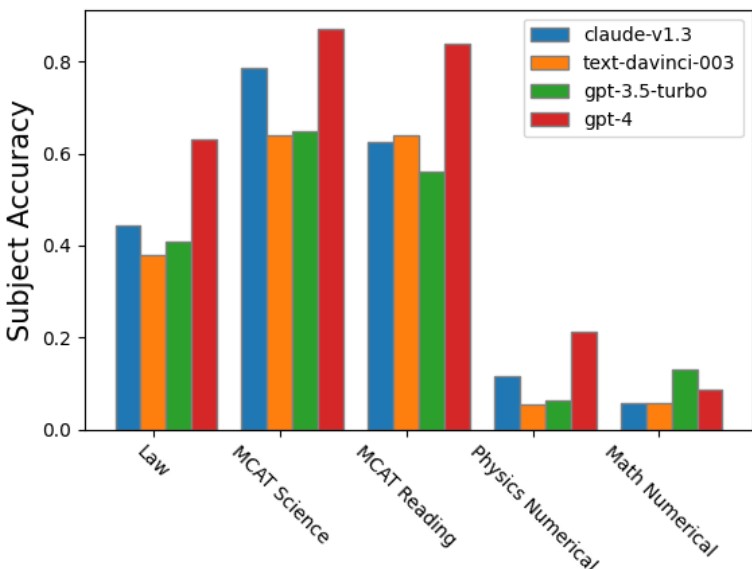

Figure 1: Accuracy of models over automatically scored components of the ARB benchmark. Numerical questions are evaluated with a relative error threshold of $10^{-2}$.

We see models generally do quite well on the multiple choice Law and MCAT subsets, but struggle significantly on questions with numerical final answers. GPT-4 is the only model capable of reliably simplifying complex expressions, but even GPT-4 struggles to reliably perform arithmetic and symbolic manipulations over long contexts.

On the multiple-choice questions, the only model that cannot reliably follow the answer formatting instructions is `gpt-3.5-turbo`. This happens for a variety of reasons, including the model refusing to answer or to commit to a single answer choice. On the Law benchmark, `gpt-3.5-turbo` does not output a parsable answer around 25% of the time. The other models exhibit this failure in less than 5% of multiple-choice questions, with GPT-4 being correctly parsed over 99% of the time.

We see a similarly low performance profile across models on symbolic problems, reported in Table 2.

Table 2: Manually parsed scores for symbolic answer questions.

|  | Math Symbolic | Physics Symbolic |
| --- | --- | --- |
| gpt-4-0314 | 15% | 20% |
| gpt-3.5-turbo-0301 | 12% | 8% |
| text-davinci-003 | 17% | 6% |
| claude-v1.3-100k | 10% | 12% |

Table 3: Mistakes on mathematics and physics problems in ARB, GPT-4.

| | Misread problem | Wrong approach | Logical error or hallucination | Arithmetic mistake | Correct answer | Correct reasoning |
|---|---|---|---|---|---|---|
| Math Numerical | 0% | 25% | 88% | 48% | 3% | 3% |
| Math Symbolic | 16% | 50% | 29% | 4% | 16% | 16% |
| Math Proof-like | 5% | 50% | 72% | 16% | n/a | 5% |
| Physics Numerical | 0% | 80% | 53% | 6% | 6% | 6% |
| Physics Symbolic | 0% | 37% | 68% | 31% | 28% | 12% |

As mentioned at the start of Section 3, benchmarks with very high scores are less useful for differentiating model capabilities. The same holds for benchmarks with very low scores across the board. On Math Numerical, GPT-4 has slightly lower accuracy than `gpt-3.5-turbo` on our run (although not with few-shot prompting, see Appendix I); similarly, `text-davinci-003` has similar accuracy as GPT-4 on Math Symbolic. After inspection, this is a combination of two factors: our dataset having several answers exactly 0 (or $\mathbb{Z}$ in cases where the answer is a group) and weaker models "guessing" correctly; and the memorization / faithful reasoning tradeoff discussed in Appendix G. Luckily, this by definition stops being an issue as models improve.

## 4.2 What Kind of Errors Do LLMs Make?

The GPT-4 evaluation paper (Bubeck et al., 2023) classified errors GPT-4 makes in single-pass evaluation on GSM8K (Cobbe et al., 2021) and MATH (Hendrycks et al., 2021) into three types: *arithmetic mistakes*, *misunderstood statement*, and *wrong approach*. We make a more fine-grained analysis and extend it to math and physics problems in our dataset. The results are in Table 3.

The errors current LLMs make on the Mathematics part of ARB fall into five general types:

- Misunderstanding / answering only a part of the question / misread problem;
- Wrong approach: the model's early chain of thought does not guess the right approach;
- Logical errors: the model uses a false implication between two statements;
- Hallucinating facts or theorems: the model confabulates a statement that is false in general, or not applicable in context;
- Arithmetic/calculation error: the model multiplies incorrectly, omits a term in an expression, gives a wrong numerical value for a fraction, and other similar mistakes.

We graded GPT-4 using the above as a guideline. Our grading of the model's CoT answers is not mutually exclusive; if the model both uses an approach that doesn't go anywhere and makes a calculation error in it, we count it towards both categories. Note that the errors might not be independent: arithmetic mistakes could be more or less frequent in wrong approach solutions as opposed to the solutions with correct idea. We notice that the model is likely to make incorrect simplifications to get to some final answer in approaches that cannot work; this is expected, as prompting the model to produce a solution with a final answer often leads it to produce *some* final answer by any means.

When the model outputs a chain of implications, it is not always clear whether some false statement is due to a logical error, or it is a straight-out confabulation. We merge those two error types in Table 3.

Some problems ask for multiple things to be proven or calculated. Our graders gave the model a score of 0.5 if it correctly derived at least half of the "subproblems" (for example, homology groups of a given manifold). With this more benevolent form of grading, the performance of GPT-4 on the Proof-like problems jumps to 16%. Where applicable, slight discrepancy with automatic evaluation is also possible due to the error tolerance. It is possible that our graders underestimate the rate of arithmetic mistakes in some cases, especially when the approach is clearly wrong, or it is not clear whether a given error is due to faulty reasoning or due to a missed term in the calculations.

We note that many of the problems in Physics Symbolic have correct symbolic answers even when there are flaws in the chain of thought reasoning of GPT-4. This is likely due to some kind of memorization, although not necessarily from the same sources: see Table 13 for an example.

The distribution of problems might be representative only of a subset of the entire dataset, because the grading was done before the dataset was finalized; the problems added later are tagged as *"additional"* in the dataset entries. For the Symbolic and Numerical subsets (see Table 1), we subsample the problems to between 20 and 40 per subject area to minimize human grading effort. this is enough for a ballpark estimate of the frequency of different errors, and is not worth increasing because attributing error types is inherently fuzzy.

## 5 MODEL-BASED RUBRIC EVALUATION

As reasoning tasks increase in complexity, it gets harder to evaluate model performance. Symbolic final answers are in some cases difficult to grade automatically. Further, we are often more interested in the correctness of the reasoning used to produce the final answer; but evaluating intermediate reasoning steps requires expert human supervision. An ideal solution would be to use LLMs as evaluators based on a reference solution; unfortunately, there are major reliability issues.

To improve reliability, we proposed generating *rubrics* as an important component of the evaluation process. The model generates the rubric from the reference solution, then evaluates any solution based on the generated rubric. To aid rubric generation, we give few-shot examples of human-written rubrics to the rubric-generating model run. We study this approach by conducting a human evaluation of GPT-4 generated rubrics and the GPT-4 grading of its own solutions using the generated rubrics.

We rated the quality of GPT-4 generated rubrics by hand and provided the results in the first two rows of Table 4. Likert scores from 1-5 are assigned to both the *coverage* of the rubric, i.e. how well it captures key subproblems, and the point breakdown. Rubric quality scores are reported in Table 5 for symbolic and proof-like problems. We find GPT-4 designs rubrics which cover the crucial solution steps well, but struggles to properly allocate points to each step based on relative importance. However, it is much better than GPT-3.5-turbo, which tends to over-allocate points to only one or two solution steps.

Table 4: Evaluations of rubric quality and GPT-4 rubric evaluation failure cases. Rubric coverage and rubric point spread are on a 1-5 Likert scale. Alternative solutions is the percentage of correct solutions found not covered by the rubric. Extra/reduced credit track how often GPT-4 erroneously assigns or deducts points. Hallucinated rubric tracks how often GPT-4 assigns points by referring to a rubric item not actually present in the rubric.

|                      | Physics Symbolic | Math Symbolic | Proof-like |
|----------------------|------------------|---------------|------------|
| Rubric coverage      | 4.42             | 4.26          | 3.94       |
| Rubric point spread  | 4.16             | 4.00          | 4.06       |
| Alternative solutions| 5%               | 2%            | 0%         |
| Extra credit         | 27%              | 18%           | 40%        |
| Reduced credit       | 11%              | 12%           | 5%         |
| Hallucinated rubric  | 0%               | 15%           | 0%         |

The obvious limitation of rubric scoring is the case of correct solutions not covered by the rubric. We find that on our benchmark, GPT-4 rarely generates a fully or even mostly partially correct solution that does not follow the rubric. Once we finished rating the model-generated rubrics, we manually graded GPT-4's solutions according to each rubric and compared the results to GPT-4's evaluation. We also annotated, for each problem, both whether GPT-4 assigned credit inappropriately or failed to assign credit when it should.

We find a moderately high correlation between GPT-4's evaluation score and the manual score. In some cases, the model, assigns an extra point or two when compared to the annotated rubric score. However, the self-eval score almost never deviates more than two points from the ground truth. The main failure mode we detect is the assignment of partial credit to attempted solutions completely outside the problem rubric, where the human evaluation score is always zero. Taken together, we believe these results suggest that rubric-based evaluation is a promising automated evaluation method.

Table 5: Average scores (out of 10 points) when assigned by human annotators versus GPT-4. Correlation is the Pearson correlation coefficient between the two scores, over all problems.

|  | Physics Symbolic | Math Symbolic | Proof-like |
|---|---|---|---|
| Human eval score | 5.00 | 3.13 | 2.65 |
| Model eval score | 5.05 | 3.37 | 3.8 |
| Correlation | 0.91 | 0.78 | 0.82 |

Having established rubric-based evaluation as a (imperfect) proxy for correctness, we now comment on the GPT-4 performance graded by the rubric. Table 5 shows GPT-4 is best at generating correct intermediate reasoning steps for physics questions. Inspecting the model outputs suggests that GPT-4 is good at recalling relevant and useful concepts in physics for solving the relevant problem; however, it can struggle with the mathematical manipulations required to solve the problem. The model is worse at recognizing the correct concepts and formulating an appropriate plan for the math questions, particularly for proof-like problems.

## 6 LIMITATIONS AND CONCLUSION

In this paper, we have presented ARB, a novel benchmark for evaluating advanced reasoning capabilities in large language models. Our dataset is composed of various problems from the sciences and law, sourced from graduate-level exams and professional resources. Despite advancements in current LLMs, their performance remains very low on the quantitative subjects, in ARB's tasks. We also introduced a rubric-based self-evaluation method, enabling LLMs to grade their own reasoning. This method is not yet reliable enough to replace human grading. We hope that this method can be further developed for more reliable and cost-effective testing of complex model outputs..

As with all other benchmarks that are not created anew and kept secret, it is possible there is data contamination. For example, the MCAT books are not available for free in most jurisdictions, but it certainly possible that some model creators have trained on it anyway.

Finally, the benchmark does not remotely cover all aspects of human ability; a model solving this benchmark perfectly could still be much worse than most educated people in many aspects. Nevertheless, we hope that increasing the difficulty standards helps the research community ground the performance of increasingly powerful models more accurately.

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

## A  DATASHEET

We present the data card, following the format proposed by Pushkarna et al. (2022).

**Dataset Owners.** `Anonymized`.

Table 6: Data overview.

| Subject | Task Type | Source |
|---|---|---|
| Mathematics | Contest problems | Gelca and Andreescu (2017) Brayman and Kukush (2018) |
| | University math | Souza and Silva (2008) Harvard University (2021) Chen and Li (1998) |
| Physics | PhD qualifying exam | Zhongguo-Kexue-Jishu-Daxue (1990) |
| Law | US Law Standardized Exam | Barbri (2007) |
| MCAT | Reading comprehension | (Campbell et al., 2017) |
| MCAT | College science | (Campbell et al., 2017) |

**Dataset Overview.** See Table 6.

**Risk and Mitigation.** There is little risk associated with this dataset, as it is intended for benchmarking reasoning capabilities of models, and it is too small to be used for training.

**Example: Typical Data Point.** Each entry of the dataset consists of a problem statement and a ground truth solution, together with some metadata. Table 22, Table 10 and Table 11 include problem statements and ground truth solutions of typical data points.

**Sensitive Human Attributes.** We have not found any sensitive human attributes in our dataset.

**Data Distributions.** Table Section 3.1 shows the problem count for each subject area and answer type. Text entries (problem statement, ground truth solution, ground truth answer) for all categories are in LaTeX, although non-quantitative subjects have very few mathematical expressions.

