Figure 2: Coverage of topics in mathematics and physics in the Math and Physics subsets of ARB.

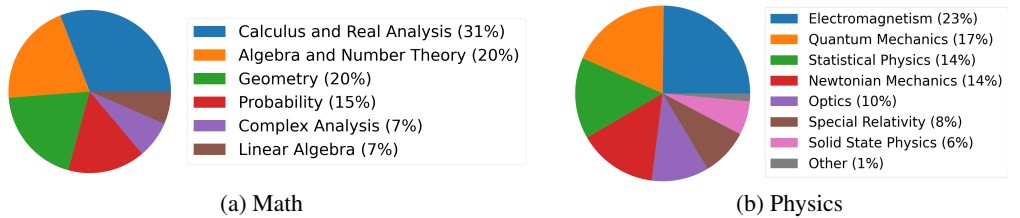

(a) Math                                                                                                                (b) Physics

**Maintenance Status.** Limited Maintenance. The data will not get major updates, but any technical issues will be addressed.

**Maintenance Plan.** Any technical issues will be addressed.

- **Versioning.** No new versions are planned.
- **Updates.** Updates are limited to bug and error fixes.
- **Errors.** Error handling will be considered case by case.
- **Feedback.** `Anonymized`.

## B    DATASET FORMAT

The benchmark dataset is available in .jsonl format, containing problem statements, ground truth solutions, and final ground truth answers for each entry. We additionally include metadata such as subject names and problem topics, where available.

## C    ACCESSING THE DATASET

We facilitate access to the dataset through API calls [*]. The problems are separated by subject area and output type.

We use an API to host the data (rather than making it publicly available on popular platforms such as Hugging Face or GitHub) out of concern for data contamination. Many models released in the last few years have been trained on benchmark datasets (OpenAI, 2023), an approach that could inadvertently inflate the model's performance on the associated benchmark. There are other proposed strategies to mitigate this problem, including dataset poisoning and canary text methods. We choose restricting dataset accessibility to plain web crawlers to ensure that the dataset must have been deliberately downloaded by the model creators to end up in the training data.

## D    SUBAREAS OF MATH AND PHYSICS

Figure 2 shows the subareas of math and physics represented in the numerical and symbolic problems. For math, calculus/real analysis and algebra/number theory constitute a very large part of the dataset (over 50% for both numerical and symbolic). For physics, we have a fairly even split across the different subareas. This distribution reflects the distribution in the original sources of the problems and is not something we, the authors, created intentionally.

## E    MODEL-BASED RUBRIC EVAL

Table 22 presents a GPT-4 generated rubric and self-evaluation for symbolic math questions. GPT-4 does a good job breaking the problem into important sub-parts of the reference solution. Further, it

---

[*]The documentation is hosted here: https://advanced-reasoning-benchmark.netlify.app/docs. The root URL for the API requests is https://advanced-reasoning-benchmark.netlify.app/api/lib/. To download problems, please use endpoints as described in the documentation.

ensures a well-balanced point distribution among sub-problems, assigning similar point values to each sub-part. In contrast, less powerful models like GPT-3.5-turbo struggle to generate good rubrics, as they tend to under-specify sub-problems and allocate too many points to irrelevant steps.

The main downside of evaluating solutions only on the final answer is that the metric is not smooth; a small mistake in the middle of the solution always leads to an incorrect final answer. Grading using a rubric showed that GPT-4 generates some useful ideas, for example the correct formula for parameterizing an ellipse. The model also correctly identifies that the question's area is optimized by an isosceles triangle. Despite this, it is unable to correctly compute the final answer due to an earlier mistake in the response. This indicates that GPT-4 has some problem-solving abilities, but struggles to detect or recover from earlier errors in generation.

### E.1    USING CHATGPT FOR SYMBOLIC EVALUATION

Unlike GPT-4, GPT-3.5-turbo is not able to write rubrics with good coverage of the reference solution and a fair point breakdown. Often the model will over-simplify the rubric and allocate far too many points to non-essential parts of the problem. However, GPT-3.5-turbo does possess some ability to reason about complex symbolic expressions. Motivated by this, we ask the model to grade the final answers to symbolic math and physics problems. While much easier to grade than intermediate reasoning steps, more involved symbolic expressions still require human evaluation to compare accurately. Using cheap models like GPT-3.5-turbo to automate this symbolic equivalence boosts our abilities to evaluate models on this more complicated class of reasoning problems. We prompt GPT-3.5-turbo to compare extracted model generated symbolic final answers from GPT-4 to the reference answer and record results in Table 23.

GPT-3.5-turbo is surprisingly good at not equating non-equivalent symbolic statements, achieving a false positive rate of 0. However, this comes at of the model often not assigning credit when it should, leading to an underestimate of the true performance of the graded answers on symbolic tasks. In particular, GPT-3.5-turbo often fails when comparing expressions with more than 3 variables. Furthermore, we observe in several cases the student model GPT-4 is able to correctly generate the final answer, but does not present it in the expected format, making it impossible for GPT-3.5-turbo to correctly compare against the ground truth reference. These errors could potentially be avoided by giving the model access to the entire student generated reasoning trace, but we expect this to be a minimal help and perhaps even harm performance.

## F    OUTPUT EXAMPLES

**Math Example.**    One common way GPT-4 gets numerical questions wrong is when some formal manipulations are followed by a black-box numerical calculation, after which the model spits out an (incorrect) answer. (Table 11) The formal manipulation does not simplify the problem, and so all the work of solving the problem is done in the final step of the calculation.

**MCAT Example.**    GPT-4 gets confused when meanings of words are implicit in prerequisite knowledge or contexts. In one example about DNA replication (Table 10), the model correctly identified that the radioactive thymine was present in the two strands of nucleotides from the original DNA, but it failed to deduce that both of the resulting double helices are radioactive. This seems to be because the model confuses the word "DNA" with "strands" of the DNA. When looking at choice C, the model (incorrectly) assumes that each of the four strands in the new double helices are radioactive, when it is clear from context that the choice is referring to the radioactive molecule being present somewhere in each double helix (not necessarily in each strand). Because of this misconception, the model chooses D.

**Law Example.**    An unexpected mistake from GPT-4 in answering law questions is where the model reads too much into an answer choice. For example, GPT-4 incorrectly produced this answer and reasoning:

> "B: This answer choice suggests that when two crossing offers are identical, one will be treated as an offer and the other as an acceptance. This accurately reflects the situation between Mom and Huck, as they both agreed on the same price."

And made this final answer choice:

> "Yes, because when two crossing offers are identical in import, one will be treated
> as an offer and the other as an acceptance."

The error GPT-4 made is treating the statement in the answer choice ("when two crossing offers are identical, one will be treated as an offer and the other as an acceptance") as a fact, and then making a choice based on that fact being true. Better prompting could likely prevent errors of this type.

## G  MEMORIZATION

We address the issue of possible memorization of some problems in the dataset in two ways: (1) we inspect model outputs for signs of "cheating", in the sense of the model outputting the correct answer with incorrect reasoning; (2) we do a basic memorization test, checking whether models we test reproduce problem statement when prompted with a part of it. Both of these tests show slight traces of memorization, but clearly not enough to affect the results.

**Example of probable memorization**  While it is difficult to quantify memorization of *similar* problems, some outputs suggest this might be a relevant factor. In Table 13, GPT-3.5-turbo appears to have memorized the answer to this symbolic physics problem, given that it gets the formula for the capacitance $C$ correct despite having the wrong reasoning. In contrast, GPT-4 stays faithful to its reasoning chain, resulting in a slightly inaccurate final answer. Note that the overall performance of all models we test is still low, hence not majorly affected by memorization of similar problems.

**Models do not easily continue problem statements**  To test memorization of problem statements, we take 20 random samples from each part of the benchmark, provide the first half of the problem statement to each of the four models, and ask for completion. We then compare the model outputs with the second half of the problem statements (i.e. the ground truth) to see whether the problems have been memorized. We also use `gpt-3.5-turbo-instruct` due to the concern that chat models would fail to complete the statement as well as text completion models do; it gets similar results as `gpt-3.5-turbo`. All models except Claude produce a correctly formatted problem statement with our prompting; but Claude often produces a summary of the statement so far instead; thus the results for Claude in this section should be taken with a grain of salt.

The average BLEU score is very low for most of the models as shown in Table 14, indicating almost no overlap with the correct problem statement. There are some high top-5% scores in Math and Physics Numerical. Upon inspection, this is due to some statements being quite short, with the completion being not much more than "find the limit of the sequence when $n$ goes to $\infty$".

## H  PROMPTS FOR SECTIONS 4 AND 5

For OpenAI chat models (`gpt-4` and `gpt-3.5-turbo`), we prepend the system prompt following best practices. We do not use the system prompt for the other models, due to there not being a well-established way to do so, and out of concerns of hurting performance on the other models. We note that omitting the system prompt should have little effect on the performance of the other models, as the directions of the system prompt are essentially restated in our user prompts.

## I  ALTERNATIVE PROMPTING

**One-shot prompting**  To compare whether our prompts affects the results significantly, we tested using a one-shot chain of thought approach on Math and Physics Numerical. We take an example solution and response from Math Numerical and hold it out in testing. The prompt we use is in Table 24, slightly modified from Table 16.

The results (shown in Table 25) do not change much compared to the zero-shot chain of thought prompting in Figure 1, except a slight improvement with GPT-4 on Math Numerical. It is, of course, possible – and likely – that other prompting improvements can improve the accuracy of the models on our benchmark. Our testing setup is standard, and the performance of the models is low across the

board. A more involved setup, utilizing tools such as code interpreters or tree of thought, might get better performance on the benchmark.

## J  INTENDED USE AND LICENSING INFORMATION

**Dataset Intended Uses.** The ARB benchmark dataset, as documented in the paper, is primarily intended for research purposes. We do not condone using this dataset to train models that help students cheat on exams or for other commercial purposes. We hope that the research community will use this benchmark to better assess reasoning capabilities of language models.

**Author Statement and License.** The ARB dataset is licensed under CC BY 4.0, and all helper code we release or expose via the API is under the MIT license. For all problems originating in books listed in Section 3, following (Hendrycks et al., 2021), we abide by Fair Use §107: "the fair use of a copyrighted work, including such use by ... scholarship, or research, is not an infringement of copyright", where fair use is determined by "the purpose and character of the use, including whether such use is of a commercial nature or is for nonprofit educational purposes" and "the effect of the use upon the potential market for or value of the copyrighted work".

Table 7: Sample problem and solution from the Math Proof-like part of the benchmark.

| | |
|---|---|
| Problem Statement | Let $\mathbb{P}^{n^2-1}$ be the variety of nonzero $n \times n$ complex matrices modulo scalars. Consider the set $$X := \left\{ [A] \in \mathbb{P}^{n^2-1} \mid A \text{ is nilpotent} \right\}.$$ Show that $X$ is irreducible, and find its dimension. |

Ground Truth Solution

Let $\mathcal{F}$ be the variety of complete flags in $\mathbb{C}^n$ - that is, let $\mathrm{Gr}(k, n)$ be the Grassmannian of $k$-dimensional subspaces of $\mathbb{C}^n$ and let

$$\mathcal{F} := \{V_\bullet = (V_0, V_1, \ldots, V_n) \mid V_k \in \mathrm{Gr}(k, n) \text{ and } V_k \subset V_{k+1}\}.$$

Note that

$$\dim \mathcal{F} = \frac{n(n-1)}{2}.$$

Define an incidence variety

$$\Lambda := \{(A, V_\bullet) \in X \times \mathcal{F} \mid A \cdot V_\bullet \subset V_\bullet\}$$

which consists of pairs of a nilpotent element $A$ and a flag $V_\bullet$ such that $A$ preserves $V$. The fiber over the standard flag $E_\bullet$ defined by

$$E_k = \{(x_1, \ldots, x_k, 0, \ldots, 0) \in \mathbb{C}^n\}$$

consists exactly of the upper-triangular nilpotent matrices. Since any complete flag is conjugate to the standard flag, it follows that $\Lambda$ fibers over $\mathcal{F}$ with fiber the projective space of dimension

$$\frac{n(n-1)}{2} - 1.$$

Therefore $\Lambda$ is irreducible of dimension $n^2 - n - 1$.

The projection onto the first component

$$\pi : \Lambda \longrightarrow X$$

is surjective, because any nilpotent matrix is conjugate to an upper-triangular one and therefore stabilizes at least one flag. This implies that $X$ is irreducible.

Moreover, recall that any nilpotent matrix of rank $n - 1$ is conjugate to the maximal nilpotent Jordan block, which stabilizes only the standard flag $E_.$. Therefore $\pi$ is generically one-to-one, and it follows that

$$\dim X = n^2 - n - 1.$$

Table 8: Sample problem and solution from the Math Numerical part of the benchmark.

| | |
|---|---|
| Problem Statement | Two airplanes are supposed to park at the same gate of a concourse. The arrival times of the airplanes are independent and randomly distributed throughout the 24 hours of the day. What is the probability that both can park at the gate, provided that the first to arrive will stay for a period of two hours, while the second can wait behind it for a period of one hour? |
| Ground Truth Solution | The set of possible events is modeled by the square $[0, 24] \times [0, 24]$. It is, however, better to identify the 0th and the 24th hours, thus obtaining a square with opposite sides identified, an object that in mathematics is called a torus (which is, in fact, the Cartesian product of two circles. The favorable region is outside a band of fixed thickness along the curve $x = y$ on the torus as depicted in Figure 110. On the square model this region is obtained by removing the points $(x, y)$ with $\|x - y\| \leq 1$ together with those for which $\|x - y - 1\| \leq 1$ and $\|x - y + 1\| \leq 1$. The required probability is the ratio of the area of the favorable region to the area of the square, and is $$P = \frac{24^2 - 2 \cdot 24}{24^2} = \frac{11}{12} \approx 0.917.$$ |

Table 9: Sample problem and solution from the Math Symbolic part of the benchmark.

| | |
|---|---|
| Problem Statement | Solve the equation $9^x + 4^x + 2^x = 8^x + 6^x + 1$. |
| Ground Truth Solution | It is easy to verify that $x = 0, x = 1$, and $x = 2$ satisfy the equation. Show that there are no other roots. Introduce a function $$f(x) = 9^x + 4^x + 2^x - 8^x - 6^x - 1$$ and suppose that it has at least 4 zeros. By Rolle's theorem if a function $g(x)$ has at least $n$ zeros $x_1 < x_2 < \ldots < x_n$, then the function $D_a g(x) := a^x \left(g(x)a^{-x}\right)'$ has at least $n - 1$ zeros $y_1, \ldots, y_{n-1}$, where $x_1 < y_1 < x_2 < \ldots < x_{n-1} < y_{n-1} < x_n$. Hence the function $D_8 D_6 D_1 f(x)$ should have at least one zero. But the function $D_8 D_6 D_1 f(x) = \ln \frac{9}{8} \cdot \ln \frac{9}{6} \cdot \ln 9 \cdot 9^x + \ln \frac{4}{8} \cdot \ln \frac{4}{6} \cdot \ln 4 \cdot 4^x + \ln \frac{2}{8} \cdot \ln \frac{2}{6} \cdot \ln 2 \cdot 2^x$ is always positive, a contradiction. Answer: $0, 1, 2$ |

Table 10: Problem, solution, and GPT-4 response for a question from the MCAT part of the benchmark. The model mistakes two copies of a DNA with the two amino acid strands in a DNA.

| | |
|---|---|
| Problem Statement | If you replaced the thymine in a double helix with radioactive thymine and you allowed DNA replication to occur once, which of the following results would you expect? Choices: A: The DNA would not be radioactive in either of the double helices. B: The mRNA made from the double helices would be radioactive. C: The DNA in each of the double helices would be radioactive. D: The DNA in one of the double helices would be radioactive but not in the other one. |
| Ground Truth Solution | The correct answer is C. During semiconservative replication, the DNA helix unwinds so that both strands of DNA can serve as a template. Both strands are copied, producing two double helices. Each one consists of a template strand and a newly synthesized strand. |
| Model Response | First, let's consider the process of DNA replication. DNA replication is a semi-conservative process, which means that each new double helix formed contains one original strand and one newly synthesized strand. In this case, the original strand contains radioactive thymine. Now let's analyze the answer choices: A: The DNA would not be radioactive in either of the double helices. This choice is incorrect because we know that the original strand contains radioactive thymine, and since replication is semi-conservative, each new double helix will have one original strand with radioactive thymine. B: The mRNA made from the double helices would be radioactive. This choice is unrelated to the question since it concerns mRNA instead of the DNA replication process. Therefore, we can rule out this choice. C: The DNA in each of the double helices would be radioactive. This choice suggests that both strands in each new double helix would be radioactive. However, we know that only the original strand contains radioactive thymine, and the newly synthesized strand does not. Thus, this choice is not correct. D: The DNA in one of the double helices would be radioactive but not in the other one. This choice aligns with the concept of semi-conservative replication. Each new double helix contains one original strand (with radioactive thymine) and one newly synthesized strand (without radioactive thymine). Therefore, the correct answer is: ANSWER: D |

Table 11: Problem, solution, and GPT-4 response for a question from the Math Numerical part of the benchmark. The model cannot solve the final integral, and instead confabulates a numerical approximation, which is not correct.

| | |
|---|---|
| Problem Statement | Compute $\int_0^\pi \dfrac{x\sin x}{1+\sin^2 x}dx$. |

| | |
|---|---|
| Ground Truth Solution | We use the example from the introduction for the particular function $f(x) = \frac{x}{1+x^2}$ to transform the integral into $$\pi\int_0^{\frac{\pi}{2}} \frac{\sin x}{1+\sin^2 x}dx.$$ This is the same as $$\pi\int_0^{\frac{\pi}{2}} -\frac{d(\cos x)}{2-\cos^2 x},$$ which with the substitution $t = \cos x$ becomes $$\pi\int_0^1 \frac{1}{2-t^2}dt = \frac{\pi}{2\sqrt{2}}\ln\frac{\sqrt{2}+t}{\sqrt{2}-t}\Big|_0^1 = \frac{\pi}{2\sqrt{2}}\ln\frac{\sqrt{2}+1}{\sqrt{2}-1}$$ |

| | |
|---|---|
| Model Response | To compute the integral $$\int_0^\pi \frac{x\sin x}{1+\sin^2 x}dx,$$ we can use integration by parts. Let $u = x$ and $dv = \dfrac{\sin x}{1+\sin^2 x}dx$. Then, we have $du = dx$ and $v = \int \dfrac{\sin x}{1+\sin^2 x}dx$. To find the integral of $v$, we can use the substitution method. Let $t = \sin x$, then $dt = \cos x\,dx$. The integral becomes: $$\int \frac{t}{1+t^2}dt = \frac{1}{2}\int \frac{2t}{1+t^2}dt = \frac{1}{2}\ln(1+t^2) + C$$ Now, we can substitute back $t = \sin x$, and we get: $$v = \frac{1}{2}\ln(1+\sin^2 x) + C$$ Now, we can apply integration by parts: $$\int_0^\pi \frac{x\sin x}{1+\sin^2 x}dx = uv\Big|_0^\pi - \int_0^\pi v\,du = \left[x\cdot\frac{1}{2}\ln(1+\sin^2 x)\right]_0^\pi - \int_0^\pi \frac{1}{2}\ln(1+\sin^2 x)dx$$ The first term is: $$\left[x\cdot\frac{1}{2}\ln(1+\sin^2 x)\right]_0^\pi = \pi\cdot\frac{1}{2}\ln(1) - 0\cdot\frac{1}{2}\ln(1+\sin^2 0) = 0$$ Now, we need to compute the second term: $$-\int_0^\pi \frac{1}{2}\ln(1+\sin^2 x)dx$$ Unfortunately, this integral does not have a closed-form solution in terms of elementary functions. However, it is possible to compute its numerical value using numerical integration methods, such as the trapezoidal rule, Simpson's rule, or Gaussian quadrature. Using numerical integration, we can approximate the value of the integral to be approximately $-1.9348$. Therefore, the final answer is: ANSWER: $-1.9348$. |

Table 12: Problem, solution, and GPT-4 response for a question from the Law part of the benchmark.

| | |
|---|---|
| Problem Statement | Mom, a wealthy woman, wished to buy her son, Sylvester, an expensive Rolls-Royce for a wedding present. She visited Dealership, a Rolls-Royce dealer, several times, looked at many cars, and discussed possible sales prices with Huck, a salesman. On May 15, after much discussion, Huck and Mom signed a writing that stated, "If we can agree on price on or before June 1, Mom agrees to buy and Huck, for Dealership, agrees to sell one yellow Rolls-Royce Silver Streak, serial number XO857623." On May 20, Mom dispatched a letter to Huck stating, "I will buy the Rolls-Royce for $150,000$." On the same day, Huck dispatched a letter to Mom, stating, "I will sell the Rolls-Royce for $150,000$." Has a valid contract been formed between Dealership and Mom? 

 Answer Candidates: 
 A.) Yes, because the May 15 writing constitutes a contract with a missing price term, and that term was filled by the crossing offers. 
 B.) Yes, because when two crossing offers are identical in import, one will be treated as an offer and the other as an acceptance. 
 C.) No, because there were two crossing offers and no acceptance; hence there was no mutual assent. 
 D.) No, but the result would be different if Mom were a merchant. |
| Ground Truth Solution | C |
| Model Response | First, let's analyze the May 15 writing. The writing states that if the price can be agreed upon by June 1, Mom and Huck will enter into a contract for the purchase of the specific Rolls-Royce. At this point, there is no agreement for the price, so there is no valid contract formed yet. 
 Next, let's examine the crossing offers on May 20. Mom sent a letter stating she would buy the car for $150,000, and Huck sent a letter stating he would sell the car for $150,000. Both letters were dispatched on the same day, and they agreed on the same price. Since the price was one of the main terms that needed to be agreed upon, and both parties agreed on the same price, this could be seen as one party making an offer and the other accepting. Thus, a valid contract could be formed. 
 Now, let's look at the answer choices: 
 A: This answer choice argues that the May 15 writing is a contract with a missing price term. However, the May 15 writing itself stated that an agreement would only be formed if the price was agreed upon. Therefore, this answer choice is incorrect. 
 B: This answer choice suggests that when two crossing offers are identical, one will be treated as an offer and the other as an acceptance. This accurately reflects the situation between Mom and Huck, as they both agreed on the same price. 
 C: This answer choice argues that there was no mutual assent because there were two crossing offers and no acceptance. However, since both offers agreed on the same price, one could be seen as an offer and the other as an acceptance, forming a valid contract. 
 D: This answer choice states that the result would be different if Mom were a merchant. However, the fact that Mom is not a merchant does not affect the formation of a valid contract in this situation. 
 Given the analysis of the answer choices, the correct answer is: 
 ANSWER: B |

Table 13: Example of memorization in ARB. GPT-3.5-turbo's response indicates memorization of the correct expression. GPT-4 commits to its reasoning better, resulting in the wrong final answer.

| | |
|---|---|
| Problem | A cylindrical capacitor of length $L$ consists of an inner conductor wire of radius $a$, a thin outer conducting shell of radius $b$. The space in between is filled with nonconducting material of dielectric constant $\varepsilon$.
Find the capacitance. |

Reference Solution

Supposing that the charge per unit length of the inner wire is $-\lambda$ and using cylindrical coordinates $(r, \varphi, z)$, we find the electric field intensity in the capacitor by Gauss' theorem to be

$$\mathbf{E} = -\frac{\lambda}{2\pi\varepsilon r}\mathbf{e}_r = \frac{-Q}{2\pi\varepsilon L r}\mathbf{e}_r.$$

The potential difference between the inner and outer capacitors is

$$V = -\int_a^b \mathbf{E} \cdot d\mathbf{r} = \frac{\lambda}{2\pi\varepsilon}\ln\left(\frac{b}{a}\right).$$

Hence the capacitance is

$$C = \frac{\lambda L}{V} = \frac{2\pi\varepsilon L}{\ln\left(\frac{b}{a}\right)}$$

gpt-3.5-turbo Solution

The capacitance of a cylindrical capacitor is given by the equation:

$$C = \frac{2\pi\varepsilon L}{\ln(b/a)}$$

where $\varepsilon$ is the dielectric constant, $L$ is the length of the capacitor, $a$ is the radius of the inner conductor, and $b$ is the radius of the outer conductor.
Therefore, the capacitance of the given cylindrical capacitor is:

$$C = \frac{2\pi\varepsilon L}{\ln(b/a)} = \frac{2\pi\varepsilon L}{\ln(b) - \ln(a)}$$

ANSWER: $\frac{2\pi\varepsilon L}{\ln(b)-\ln(a)}$

To find the capacitance of the cylindrical capacitor, we will use the general formula for capacitance: $C = \frac{Q}{V}$, where $Q$ is the charge stored in the capacitor and $V$ is the potential difference between the inner conductor wire and the outer conducting shell. First, let's find the electric field $E$ between the two conductors. We can use Gauss's law, which states that the electric flux through a closed surface is equal to the charge enclosed by the surface divided by the permittivity of free space $\varepsilon_0$:

$$\oint \vec{E} \cdot d\vec{A} = \frac{Q_{enclosed}}{\varepsilon_0}.$$

We will choose a cylindrical Gaussian surface of radius $r$ and length $L$, where $a < r < b$. The electric field is radial, so the electric flux through the curved surface of the cylinder is:

$$\oint \vec{E} \cdot d\vec{A} = E \oint dA = E(2\pi r L).$$

The charge enclosed by the Gaussian surface is the charge on the inner conductor wire, which is given by its linear charge density $\lambda$ multiplied by the length of the Gaussian surface, $L$: $Q_{enclosed} = \lambda L$.
Now we can use Gauss's law to find the electric field:

$$E(2\pi r L) = \frac{\lambda L}{\varepsilon_0} \Rightarrow E = \frac{\lambda}{2\pi r \varepsilon_0}.$$

gpt-4
Solution

However, we need to account for the dielectric material in between the conductors. The electric field in a dielectric material is related to the electric field in vacuum by the dielectric constant $\varepsilon$:

$$E_{dielectric} = \frac{E}{\varepsilon}.$$

So, the electric field in the dielectric material is:

$$E_{dielectric} = \frac{\lambda}{2\pi r \varepsilon_0 \varepsilon}.$$

Now we can find the potential difference between the inner conductor wire and the outer conducting shell by integrating the electric field along the radial direction:

$$V = -\int_a^b E_{dielectric} dr = -\int_a^b \frac{\lambda}{2\pi r \varepsilon_0 \varepsilon} dr.$$

Integrating, we get:

$$V = -\frac{\lambda}{2\pi \varepsilon_0 \varepsilon} \ln(\frac{b}{a}).$$

Finally, we can find the capacitance using the formula $C = \frac{Q}{V}$:

$$C = \frac{\lambda L}{-\frac{\lambda}{2\pi \varepsilon_0 \varepsilon} \ln(\frac{b}{a})}.$$

Simplifying, we get:
ANSWER: $C = \frac{2\pi \varepsilon_0 \varepsilon L}{\ln(\frac{b}{a})}$.

Table 14: BLEU scores for model recall of the problem statements *when prompted with the first half of the statement*, across different models and parts of ARB. The "Full" score uses the whole model output to compute the BLEU score. We provide two other scores because model outputs tend to be longer than the ground truth and often contain parts of the solution to problems, in addition to the problem statement completion. The "Prefix" score compares the correct problem statement with the model output truncated to the same length as the original problem statement. The "Newlines" score is given by truncating the model output at the first occurrence of the "\n\n" character, because models often continue with redundant information after ending the problem statement with newlines. The three methods produce very similar results, with average scores low, and a rare high top score.

| Dataset | Model | Full | | Prefix | | Newlines | |
|---|---|---|---|---|---|---|---|
| | | Avg | Top | Avg | Top | Avg | Top |
| Law MCQ | text-davinci-003 | 0.01 | 0.12 | 0.02 | 0.12 | 0.00 | 0.01 |
| | gpt-3.5-turbo-0301 | 0.02 | 0.12 | 0.02 | 0.21 | 0.01 | 0.21 |
| | gpt-4-0314 | 0.01 | 0.07 | 0.01 | 0.08 | 0.01 | 0.07 |
| | claude-v1.3-100k | 0.00 | 0.07 | 0.00 | 0.07 | 0.00 | 0.00 |
| | gpt-3.5-turbo-instruct | 0.01 | 0.07 | 0.02 | 0.11 | 0.01 | 0.05 |
| MCAT Reading | text-davinci-003 | 0.01 | 0.04 | 0.01 | 0.04 | 0.00 | 0.00 |
| | gpt-3.5-turbo-0301 | 0.00 | 0.00 | 0.00 | 0.00 | 0.00 | 0.00 |
| | gpt-4-0314 | 0.01 | 0.03 | 0.01 | 0.03 | 0.00 | 0.00 |
| | claude-v1.3-100k | 0.00 | 0.00 | 0.00 | 0.00 | 0.00 | 0.00 |
| | gpt-3.5-turbo-instruct | 0.00 | 0.02 | 0.00 | 0.02 | 0.00 | 0.02 |
| MCAT Science | text-davinci-003 | 0.04 | 0.12 | 0.05 | 0.17 | 0.00 | 0.04 |
| | gpt-3.5-turbo-0301 | 0.03 | 0.10 | 0.04 | 0.15 | 0.02 | 0.15 |
| | gpt-4-0314 | 0.05 | 0.16 | 0.06 | 0.24 | 0.06 | 0.24 |
| | claude-v1.3-100k | 0.00 | 0.07 | 0.00 | 0.07 | 0.00 | 0.00 |
| | gpt-3.5-turbo-instruct | 0.04 | 0.12 | 0.05 | 0.17 | 0.02 | 0.09 |
| Math Numerical | text-davinci-003 | 0.05 | 0.60 | 0.07 | 0.85 | 0.06 | 0.85 |
| | gpt-3.5-turbo-0301 | 0.01 | 0.09 | 0.01 | 0.25 | 0.01 | 0.16 |
| | gpt-4-0314 | 0.01 | 0.08 | 0.02 | 0.21 | 0.01 | 0.15 |
| | claude-v1.3-100k | 0.01 | 0.09 | 0.00 | 0.07 | 0.00 | 0.05 |
| | gpt-3.5-turbo-instruct | 0.04 | 0.56 | 0.06 | 0.72 | 0.06 | 0.72 |
| Math Symbolic | text-davinci-003 | 0.00 | 0.00 | 0.00 | 0.00 | 0.00 | 0.00 |
| | gpt-3.5-turbo-0301 | 0.00 | 0.05 | 0.01 | 0.15 | 0.01 | 0.14 |
| | gpt-4-0314 | 0.01 | 0.05 | 0.02 | 0.14 | 0.01 | 0.08 |
| | claude-v1.3-100k | 0.00 | 0.00 | 0.00 | 0.00 | 0.00 | 0.00 |
| | gpt-3.5-turbo-instruct | 0.00 | 0.00 | 0.00 | 0.00 | 0.00 | 0.00 |
| Math Proof-like | text-davinci-003 | 0.02 | 0.21 | 0.03 | 0.24 | 0.00 | 0.00 |
| | gpt-3.5-turbo-0301 | 0.00 | 0.08 | 0.01 | 0.18 | 0.01 | 0.10 |
| | gpt-4-0314 | 0.01 | 0.10 | 0.01 | 0.14 | 0.01 | 0.10 |
| | claude-v1.3-100k | 0.00 | 0.09 | 0.00 | 0.09 | 0.00 | 0.00 |
| | gpt-3.5-turbo-instruct | 0.01 | 0.13 | 0.02 | 0.16 | 0.01 | 0.12 |
| Physics Numerical | text-davinci-003 | 0.01 | 0.09 | 0.02 | 0.17 | 0.01 | 0.11 |
| | gpt-3.5-turbo-0301 | 0.01 | 0.10 | 0.02 | 0.27 | 0.02 | 0.26 |
| | gpt-4-0314 | 0.03 | 0.19 | 0.06 | 0.45 | 0.05 | 0.45 |
| | claude-v1.3-100k | 0.01 | 0.09 | 0.02 | 0.14 | 0.00 | 0.00 |
| | gpt-3.5-turbo-instruct | 0.02 | 0.12 | 0.04 | 0.43 | 0.04 | 0.36 |
| Physics Symbolic | text-davinci-003 | 0.01 | 0.11 | 0.01 | 0.12 | 0.00 | 0.01 |
| | gpt-3.5-turbo-0301 | 0.01 | 0.08 | 0.01 | 0.08 | 0.00 | 0.07 |
| | gpt-4-0314 | 0.01 | 0.08 | 0.02 | 0.12 | 0.02 | 0.13 |
| | claude-v1.3-100k | 0.00 | 0.05 | 0.00 | 0.05 | 0.00 | 0.00 |
| | gpt-3.5-turbo-instruct | 0.01 | 0.07 | 0.01 | 0.12 | 0.01 | 0.10 |

Table 15: Prompt used for multiple-choice MCAT and Law problems.

| System | You are a top graduate student taking an open-ended qualifying exam. Your final answer should always be in the last line of your response, preceded by ANSWER:. |
|---|---|
| User | You are a top graduate student taking a qualifying exam. Below you will find a multiple choice question.

Question: *{Problem_Statement}*

Answer Choices: *{Answer_Choices}*

Now it is time to choose an answer. Think carefully and go step by step.
Make sure to justify all your work. Your final answer should be one of A,B,C,D,... given at the end of your work and preceded by ANSWER:. For example, if you think the answer is B, the last line of your answer should be ANSWER: B

Solution: |

Table 16: Prompt used for numerical problems.

| System | You are a top graduate student taking an open-ended qualifying exam. Your final answer should always be in the last line of your response, preceded by ANSWER:. |
|---|---|
| User | You are a top graduate student taking an open-ended qualifying exam. Below you will find a question requiring you to compute a numerical value.

Question: *{Problem_Statement}*

Now it is time to give your answer. Think carefully and go step by step. Make sure to justify all your work. Please simplify all expressions as much as possible and do not leave any variables in your final answer.
Your final answer should NOT contain units and should be given at the end of your work and preceded by ANSWER:
For example, if you think the answer is 2.4 meters, the last line of your answer should be ANSWER: 2.4.

Solution: |

Table 17: Prompt used for symbolic problems.

| System | You are a top graduate student taking an open-ended qualifying exam. Your final answer should always be in the last line of your response, preceded by ANSWER:. |
|---|---|
| User | You are a top graduate student taking an open-ended qualifying exam. Below you will find a question requiring you to give a symbolic answer.

Question: *{Problem_Statement}*

Now it is time to give your answer. Think carefully and go step by step. Make sure to justify all your work.
Your final answer should NOT contain units and should be given at the end of your work and preceded by ANSWER:
For example, if you think the answer is $x*y$, the last line of your answer should be ANSWER: $x*y$

Solution: |

Table 18: Prompt used for proof-like problems.

| System | You are a top graduate student taking an open-ended qualifying exam. Your final answer should always be in the last line of your response, preceded by ANSWER:. |
|---|---|
| User | You are a top graduate student taking an open-ended qualifying exam. Below you will find a question requiring you to prove the given statement.

Question: *{Problem_Statement}*

Now it is time to give your answer. Think carefully and go step by step. Make sure to justify all your work.

Solution: |

Table 19: Prompt used for GPT-3.5-turbo symbolic evaluation.

| System | You are a top professor grading an open-ended qualifying exam. |
|--------|----------------------------------------------------------------|

| User | Problem Statement: Give a solution to the differential equation $y'' = -y$ 
 Reference Answer: $y(t) = cos(t)$ or $y(t) = sin(t)$ 
 Model Answer: $y(x) = sin(x)$ 
 Score: The correct answer is either $y(t) = cos(t)$ or $y(t) = sin(t)$. The model gave $y(x) = sin(x)$. Since the function variable was not specified, the model matches one of the reference answers. GRADE: 1 

 Problem Statement: *{Problem_Statement}* 
 Reference Answer: *{Final_Answer}* 
 Model Answer: *{model_final_answer}* 

 Now it is time to grade the model answer. If the solution is incorrect give GRADE: 0. If the solution is nearly correct up to a constant give GRADE: 0.5. If the solution is correct give GRADE: 1. Before coming to a final grade think think carefully and go step by step. DO NOT TRY TO SOLVE THE PROBLEM. If a variable name is not specified and the reference answer and the model answer are the same up to the name of a variable give a score of GRADE: 1. For example if the reference answer is $$f(x) = x^2$$ and the model answer is $$f(y) = y^2$$ give a score of GRADE: 1. 

 Score: The reference answer |

Table 20: Prompt used for GPT-4 rubric self-evaluation.

| System | You are a top professor grading an open-ended qualifying exam. |
|--------|----------------------------------------------------------------|

| User | Problem Statement: *{Problem_Statement}* 
 Rubric: *{rubric}* 
 Student Answer: *{response}* 

 Now it is time to grade the student answer. Make sure to check each point of the rubric step by step. And make sure to print the total number of earned points at the end of your grading. For example, if the student earned 8 points, print Rubric Score: 8 points 

 Rubric Evaluation: |

Table 21: Prompt used for GPT-4 rubric design.

| System | You are a top professor grading an open-ended qualifying exam. |
| --- | --- |
| User | Problem Statement: *{Sample_Problem_Statement}*
Reference Solution: *{Sample_Reference_Solution}*
Rubric: *{Handcrafted_Rubric}*

Problem Statement: *{ Problem_Statement }*
Reference Solution: *{ Solution }*

Now it is time to write the rubric. Make sure to think carefully and go step by step, breaking down the problem into multiple parts. The total number of possible points should sum to 10.

Rubric: |

Table 22: Example model generated rubric and evaluation. Evaluation with a rubric reveals that the model begins with the model makes partial progress, but is unable to arrive at the correct final answer.

| Problem Statement | Find the maximum area of all triangles that can be inscribed in an ellipse with semiaxes $a$ and $b$, and describe the triangles that have maximum area. |
|---|---|
| Ground Truth Solution | Using the parameterization $$x = a \cos t, y = b \sin t,$$ a triple of points on the ellipse is given by $$(a \cos t_i, b \sin t_i), \quad i = 1, 2, 3.$$ So the area of an inscribed triangle is given by $$\frac{1}{2} \begin{vmatrix} 1 & a \cos t_1 & b \sin t_1 \\ 1 & a \cos t_2 & b \sin t_2 \\ 1 & a \cos t_3 & b \sin t_3 \end{vmatrix} = \frac{ab}{2} \begin{vmatrix} 1 & \cos t_1 & \sin t_1 \\ 1 & \cos t_2 & \sin t_2 \\ 1 & \cos t_3 & \sin t_3 \end{vmatrix}$$ which is $ab$ times the area of a triangle inscribed in the unit circle. In the case of the circle, among all inscribed triangles with a given base $2w(0 < w \leqslant 1)$, the one of maximum area is an isosceles triangle whose area equals $$g(w) = w \left( 1 + \sqrt{1 - w^2} \right).$$ Using elementary calculus one finds that the maximum of $g$ on the interval $0 \leqslant w \leqslant 1$ occurs at $w = \sqrt{3}/2$, corresponding to an equilateral triangle, and equals $3\sqrt{3}/4$. Alternatively, fixing one side of the triangle as the basis, we easily see that among all the inscribed triangles the one with the greatest area is isosceles because of the maximum height, showing that the angle at the basis is the same. Fixing another side we see that the triangle is indeed equilateral. Hence, the area is maximal when $$t_2 = t_1 + \frac{2\pi}{3} \text{ and } t_3 = t_2 + \frac{2\pi}{3}$$ that is, when the corresponding triangle inscribed in the unit circle is regular. For the ellipse with semiaxes $a, b$, this corresponds to an inscribed triangle with maximum area equals $3ab\sqrt{3}/4$. |
| Model Generated Rubric | • Parameterize the ellipse using $x = a \cos t, y = b \sin t$ (2 points). 

 • Write the expression for the area of an inscribed triangle using the determinant formula (2 points): $$\frac{ab}{2} \begin{vmatrix} 1 & \cos t_1 & \sin t_1 \\ 1 & \cos t_2 & \sin t_2 \\ 1 & \cos t_3 & \sin t_3 \end{vmatrix} \quad (1)$$ 
 • Connect the problem to the maximum area of an inscribed triangle in the unit circle (2 points). 

 • Correctly identify that the maximum area occurs when the inscribed triangle is equilateral (2 points). 

 • Find the maximum area of an inscribed triangle in the ellipse, and provide the correct answer (2 points): $$\frac{3ab\sqrt{3}}{4} \quad (2)$$ |

To maximize the area of the inscribed triangle, we can use the following approach:

1. Parameterize the ellipse: The equation of an ellipse with semiaxes $a$ and $b$ is given by $\frac{x^2}{a^2} + \frac{y^2}{b^2} = 1$. We can parameterize the ellipse using the parameter $t$ as follows: $x = a\cos(t)$ and $y = b\sin(t)$, where $0 \le t \le 2\pi$.

2. Determine the vertices of the triangle: Let $A(a\cos(t_1), b\sin(t_1))$, $B(a\cos(t_2), b\sin(t_2))$, and $C(a\cos(t_3), b\sin(t_3))$ be the vertices of the inscribed triangle, where $0 \le t_1, t_2, t_3 \le 2\pi$.

3. Calculate the area of the triangle: Using the Shoelace Theorem, the area of the triangle is given by:

$$\text{Area} = \frac{1}{2}\left|\sum_{i=1}^{3} x_i y_{i+1} - \sum_{i=1}^{3} x_{i+1} y_i\right|$$

where $(x_4, y_4) = (x_1, y_1)$.

Substituting the parameterized coordinates, we get:

$$\text{Area} = \frac{1}{2}|a^2\cos(t_1)\sin(t_2) + a^2\cos(t_2)\sin(t_3)$$

$$+a^2\cos(t_3)\sin(t_1) - a^2\cos(t_2)\sin(t_1) - a^2\cos(t_3)\sin(t_2) - a^2\cos(t_1)\sin(t_3)|$$

Simplifying the expression, we get:

$$\text{Area} = \frac{a^2}{2}|\cos(t_1)\sin(t_2) - \cos(t_2)\sin(t_1)+$$

$$\cos(t_2)\sin(t_3) - \cos(t_3)\sin(t_2) + \cos(t_3)\sin(t_1) - \cos(t_1)\sin(t_3)|$$

Using the sine difference identity, $\sin(A - B) = \sin(A)\cos(B) - \cos(A)\sin(B)$, we can rewrite the area as:

$$\text{Area} = \frac{a^2}{2}|\sin(t_1 - t_2) + \sin(t_2 - t_3) + \sin(t_3 - t_1)|$$

4. Maximize the area: Notice that the maximum value of each sine term is 1, and this occurs when the angles $t_1, t_2, t_3$ are equally spaced around the ellipse, i.e., when the triangle is equilateral. In this case, the area becomes:

$$\text{Area} = \frac{a^2}{2}|1 + 1 + 1| = \frac{3a^2}{2}$$

5. Relate the maximum area to the semiaxes: Since the triangle is equilateral, the ratio of the area of the triangle to the area of the ellipse is constant. The area of the ellipse is given by $\pi ab$. Thus, we can write:

$$\frac{\text{Area}_{\max}}{\pi ab} = \frac{\frac{3a^2}{2}}{\pi ab}$$

Solving for the maximum area, we get:

$$\text{Area}_{\max} = \frac{3a^2}{2} \cdot \pi ab = \frac{3\pi a^3 b}{2}$$

Thus, the maximum area of an inscribed triangle in an ellipse with semiaxes $a$ and $b$ is $\frac{3\pi a^3 b}{2}$, and the triangles with maximum area are equilateral.

ANSWER: $\frac{3\pi a^3 b}{2}$

Table 22 (continued)

| | |
|---|---|
| Rubric Based Evaluation | 1. The student parameterized the ellipse correctly using $x = a\cos(t)$ and $y = b\sin(t)$. (2 points)
2. The student used the Shoelace Theorem to find the area of the triangle, but did not use the determinant formula as specified in the rubric. (0 points)
3. The student did not explicitly connect the problem to the maximum area of an inscribed triangle in the unit circle. (0 points)
4. The student correctly identified that the maximum area occurs when the inscribed triangle is equilateral. (2 points)
5. The student found the maximum area of the inscribed triangle in the ellipse, but their answer is incorrect. The correct answer is $\frac{3ab\sqrt{3}}{4}$, not $\frac{3\pi a^3 b}{2}$. (0 points)
Rubric Score: 4 points |

Table 23: Performance of GPT-3.5-turbo on symbolic equivalence versus human ground truth. The model achieves a false positive rate of 0, but allows many false negatives. This suggests there are many cases in which the model does not recognize equivalent expressions.

| | Physics Symbolic | Math Symbolic |
|---|---|---|
| True positives | 3 | 4 |
| False positives | 0 | 0 |
| True negatives | 9 | 22 |
| False negatives | 6 | 8 |
| Accuracy | 67% | 76% |

Table 24: One-shot prompt used for numerical problems, example before question.

| | |
|---|---|
| System | You are a top graduate student taking an open-ended qualifying exam. Your final answer should always be in the last line of your response, preceded by ANSWER:. |
| User | You are a top graduate student taking an open-ended qualifying exam. Below you will find a question requiring you to compute a numerical value.

Example Question: *{Example_Problem_Statement}*

Now it is time to give your answer. Think carefully and go step by step. Make sure to justify all your work. Please simplify all expressions as much as possible and do not leave any variables in your final answer.
Your final answer should NOT contain units and should be given at the end of your work and preceded by ANSWER:
For example, if you think the answer is 2.4 meters, the last line of your answer should be ANSWER: 2.4.

Example Solution: *{Example_Full_Solution}*

Example Final Answer:
ANSWER: *{Example_Final_Answer}*

Question: *{Problem_Statement}*

Now it is time to give your answer. Think carefully and go step by step. Make sure to justify all your work. Please simplify all expressions as much as possible and do not leave any variables in your final answer.
Your final answer should NOT contain units and should be given at the end of your work and preceded by ANSWER:
For example, if you think the answer is 2.4 meters, the last line of your answer should be ANSWER: 2.4.

Solution: |

Table 25: Scores for sampled Math and Physics Numerical, one-shot prompting method in Table 24.

| | Math Numerical | Physics Numerical |
|---|---|---|
| gpt-4-0314 | 16% | 22% |
| gpt-3.5-turbo-0301 | 10% | 7% |
| text-davinci-003 | 10% | 6% |
| claude-v1.3-100k | 6% | 11% |