# OpenReview forum: "ARB: Advanced Reasoning Benchmark for Large Language Models"
_ICLR.cc/2024/Conference — Submitted to ICLR 2024_

### Official Review · Reviewer_Vtbi · 2023-10-31

**Soundness:** 3 good
**Presentation:** 3 good
**Contribution:** 3 good
**Rating:** 6
**Confidence:** 2

**Summary:**

The authors have developed a challenging benchmark that consists of questions in mathematics, physics, biology, chemistry, and law.
Comparing to the existing benchmarks, this benchmark requires expert level domain specific knowledge and reasoning.
Further, instead of the traditional multiple-choice questions form, all the problems are short-answer or open-domain questions.
For the ease of grading the test based questions, the authors uses a rubric based method, where a language models first raise rubrics according to the answer, then it grades the generated answer according to the rubric. This yields a automated grading method which is reasonably performant though not as reliable as human.

**Strengths:**

Originality: 4/5
Although there are existing benchmarks that focuses on exploiting the incapability of the existing language models, such as big-bench, this paper provides a new aspect on how to automatically grade the short answer and open-domain question answering tasks with GPT-4.

Quality: 4/5
The claims of this paper are well bolstered by experiments and data. One aspect is how well does existing language model perform on this benchmark, and another aspect is that how well does the automatic grading system perform compared to the human expert on these questions.

Clarity: 4/5
The paper is nicely written and pretty clear.

Significance: 4/5
This paper could be impactful both due to its challenging natural and its evaluation paradigm.

**Weaknesses:**

As the limitation states, this benchmark could have potentially be seen through the training dataset and lead to the contamination problem. For example, an answer only have the correct final step but not the correct intermediate step.

Further, the dataset is comparatively small comparing to the existing datasets.

**Questions:**

Among the problems, how many of the symbolic problems can be convert into concrete calculation based problems?

**Details Of Ethics Concerns:**

Will it introduce copy right issues?

---

> ### Author Response · Authors · 2023-11-23
>
> Thank you for the review; here are the answers to your questions.
>
> >  As the limitation states, this benchmark could have potentially be seen through the training dataset and lead to the contamination problem.
>
> Thank you for pointing this out. We do not have access to the training set of any of the tested models. In spite of this, we can test for memorization in this manner: we ask models to complete problem statements randomly sampled from the dataset, and compare their outputs against the true continuation of the problem statement. (Please see Appendix G for details.)
>
> As the similarity scores are quite low, we claim that models we test do not easily complete the exact problem statements, which rules out the naive memorization hypothesis.
>
> However, memorization in LLMs is a complex topic, and the fact that the text of a problem is not memorized does not necessarily mean the model has not memorized a similar problem or fact; we also discuss this in Appendix G. We are not aware of other works that successfully resolve this issue.
>
>
> > Among the problems, how many of the symbolic problems can be convert into concrete calculation based problems?
>
> Thank you for asking this. The answer to this question strongly depends on what is meant by “concrete calculation,”. However, if we were to define this to mean that the ground truth solution mostly consists of equations, then around a third of the math problems satisfy this criteria. As for the physics problems, almost all of the problems are concrete calculations (almost by nature of the subject).

---

### Official Review · Reviewer_c9vS · 2023-11-02

**Soundness:** 3 good
**Presentation:** 3 good
**Contribution:** 3 good
**Rating:** 6
**Confidence:** 3

**Summary:**

This paper proposes ARB, a new dataset for evaluating LLM reasoning in expert domains such as mathematics, physics, chemistry, biology and law. Though there is technical contribution, this is still an important contribution to the community due to the lack of good benchmarks.

**Strengths:**

This paper proposes a new dataset ARB covering a wide range of domains for reasoning.
This paper evaluates 3 common models (ChatGPT, GPT-4 and Claude) on ARB. The authors also provide a breakdown of the error cases in GPT-4, which provide insights for future directions.
This paper proposes model-based rubric evaluation. The authors rigorously verify this approach by comparing the grading of GPT-4 and humans, which shows a moderately high correlation between them. This may be used as an evaluation tool for this dataset in the future.

**Weaknesses:**

It’s not intuitive what questions are covered by ARB. Can you put a few examples in the paper?
It’s not clear what the position of ARB is compared to existing benchmarks. The authors claim ARB is more difficult. However, there is no strong supporing evidence except for the weak performance of models on the physics and math portions of ARB

**Questions:**

Section 3.1 “aspirational” -> Is it semantically correct here?
Figure 1. Can you use a higher resolution or pdf instead? The y-axis has a wrong unit.

---

> ### Author Response · Authors · 2023-11-23
>
> We are grateful for the positive review; here are the answers to your questions.
>
> > It’s not intuitive what questions are covered by ARB. Can you put a few examples in the paper?
>
> Thank you for the suggestion! We do provide examples along with model outputs in Tables 10 through 13 (numbering of the rebuttal version). We now include some more examples in Tables 7 through 9 of the Appendix.
>
> > It’s not clear what the position of ARB is compared to existing benchmarks. The authors claim ARB is more difficult.
>
> In our updated version we do a better job of placing our work in context of the very recent literature.
>
>
> The most relevant seems to be [1], which is a similar work as ours, developed concurrently.  The main differences are: (1) their quantitative problems are slightly easier than in our benchmark, and require much less deep math knowledge; (2) our math/physics questions are not multiple-choice; (3) their quantitative problems are drawn from a single source which are somewhat more likely to be thoroughly discussed online.
>
> We also cite, among others, [2], which is indeed quite challenging, but is in a different area. We moderated our language (in the paper and in the abstract) to avoid claiming that our benchmark is uniquely challenging for current models.
>
> > Writing errors
>
> Thank you, we have now fixed the “aspirational” sentence and the units in the figure.
>
>
> [1] Arora, D., & Singh, H. G. (2023). Have LLMs Advanced Enough? A Challenging Problem Solving Benchmark For Large Language Models. arXiv preprint arXiv:2305.15074.
>
>
> [2] Valmeekam, K., Olmo, A., Sreedharan, S., & Kambhampati, S. (2022). Large Language Models Still Can't Plan (A Benchmark for LLMs on Planning and Reasoning about Change). arXiv preprint arXiv:2206.10498.

---

### Official Review · Reviewer_jkxa · 2023-11-03

**Soundness:** 2 fair
**Presentation:** 2 fair
**Contribution:** 2 fair
**Rating:** 5
**Confidence:** 3

**Summary:**

This paper proposed a new benchmark, Advanced reasoning Benchmark (ARB), for evaluating advanced reasoning capabilities in large language models. The dataset is composed of various problems from the sciences and law, sourced from graduate-level exams and professional resources, and the performance of current LLMs are relatively low on this dataset comparing to other benchmarks. The paper also introduced a rubric-based self-evaluation method, enabling LLMs to grade their own reasoning, and the authors have conducted human evaluations showing some alignment between the rubric-based self-evaluation method and human preference.

**Strengths:**

[+] ARB is a novel and challenging benchmark that extends the frontier of what LLMs are currently being tested against, covering advanced topics.

[+] The mistakes analysis in Table 3 has novelty and may provide some insights into why LLMs make an mistake.

**Weaknesses:**

[-] The evaluation steps for this dataset seems quite complicated (e.g., lots of regex), and it is unclear how to conduct easy evaluation on the open-response questions.

[-] It is also unclear how the current low performance on the ARB benchmark is not due to under-claiming.

[-] The solutions to these problems sets from textbooks may already in the training data. How to deal with this situation?

**Questions:**

- How to conduct easy evaluation on the open-response questions in the ARB dataset?

- How the current low performance on the ARB benchmark is not due to under-claiming?

- How do you grade the types of mistakes GPT-4 make in Table 3? Is it human evaluation or are there some hard-coded rules.

===After rebuttal===
Thanks for the authors' response. It still seems to me that the analysis of the current dataset depends a bit heavily on human evaluation, and it is still unclear to me the the difficulty actually came from reasoning rather than problem format or previous knowledge.

---

> ### Author Response · Authors · 2023-11-23
>
> Thank you for the review and the questions! We answer your concerns one by one.
>
> > The solutions to these problems sets from textbooks may already in the training data. How to deal with this situation?
>
> Unfortunately, we do not have access to the training set for any of the tested models. In spite of this, we can test for memorization in this manner: we ask models to complete problem statements randomly sampled from the dataset, and compare their outputs against the true continuation of the problem statement. (See Appendix G for details.)
>
> As the similarity scores are quite low, we claim that models we test do not easily complete the exact problem statements, which rules out the naive memorization hypothesis.
>
> However, memorization in LLMs is a complex topic, and the fact that the text of a problem is not memorized does not necessarily mean the model has not memorized a similar problem or fact; we also discuss this in Appendix G. We are not aware of other works that successfully resolve this issue.
>
> > How do you grade the types of mistakes GPT-4 make in Table 3? Is it human evaluation or are there some hard-coded rules.
>
> It is human evaluation. We say in Section 4.2:
> "It is possible that our graders underestimate the rate of arithmetic mistakes in some cases, especially when the approach is clearly wrong, or it is not clear whether a given error is due to faulty reasoning or due to a missed term in the calculations."
>
> The graders were given the descriptions of the five general types of errors as guidance; in addition, the graders were familiar with the relevant section of [1]. We are not sure automated evaluation of detailed error types is possible using current technology; the only way to reduce noise would be to employ more graders.
>
>
> > It is also unclear how the current low performance on the ARB benchmark is not due to under-claiming.
>
> Thank you for bringing this up. We definitely agree that there is no way to prove that models cannot do this, as our prompts are surely not optimal. It is not clear to us, at this stage, what the “best” way to prompt and use models for our benchmark; and we think finding out the answer is outside the scope of this benchmark paper. We now additionally do one-shot testing on the Numerical part of the benchmark to see if anything improves, and it doesn’t.
>
>
>
> [1] Bubeck, S., Chandrasekaran, V., Eldan, R., Gehrke, J., Horvitz, E., Kamar, E., Lee, P., Lee, Y. T., Li, Y., Lundberg, S., et al. (2023). Sparks of artificial general intelligence: Early experiments with GPT-4. arXiv preprint arXiv:2303.12712.

---

### Official Review · Reviewer_Bwn3 · 2023-11-09

**Soundness:** 2 fair
**Presentation:** 3 good
**Contribution:** 2 fair
**Rating:** 5
**Confidence:** 3

**Summary:**

The paper focuses on a novel benchmark composed of advanced reasoning problems across multiple fields like mathematics, physics, biology, chemistry, and law. Its aim is to test large language models (LLMs) like GPT-4 and Claude, noting that these models score below 50% on the proposed benchmark consisting of demanding tasks.

**Strengths:**

- The paper is well written and easy to follow
- The benchmark covers a diverse set of domains and problems.
- Results show that the benchmark is sufficiently challenging for current state-of-the-art LLMs.
- The authors present a method to ease the process of human evaluation on harder problems.

**Weaknesses:**

- I somewhat disagree with the motivation behind this work, which suggests that other benchmarks are not challenging enough. For instance, [1] and [2] have shown that even the best models either cannot generate plans or solve JEE problems, respectively, with an accuracy below 50%. While I concur that there are easier benchmarks, there are also more challenging ones available. I view this work as an additional benchmark to the existing ones that involves a broader set of domains.
- The paper appears to lack a dataset quality verification process. Moreover, it’s not entirely clear how solving the benchmark (especially  MCAT or Law questions) would correlate with reasoning ability.
- There are mixed approaches for evaluation.  Multiple-choice, numerical, and the simpler symbolic questions have automated evaluation, whereas the more complex symbolic and proof-based questions require human evaluation. The model-based evaluation is potentially beneficial, but on average, it incorrectly assigns or deducts points for 37% of the questions. This dependency on expert human evaluators limits the practicality of using this benchmark.

[1] Arora, D., & Singh, H. G. (2023). Have LLMs Advanced Enough? A Challenging Problem Solving Benchmark For Large Language Models. arXiv preprint arXiv:2305.15074.

[2] Valmeekam, K., Olmo, A., Sreedharan, S., & Kambhampati, S. (2022). Large Language Models Still Can't Plan (A Benchmark for LLMs on Planning and Reasoning about Change). arXiv preprint arXiv:2206.10498.

**Questions:**

- Is a non-parsable answer deemed as an incorrect answer? Or is that question not even considered in the final evaluation?
- Given the rapid iterations of LLMs with new and more extensive training data, a diagonalization procedure could be worth investigating, as most of the benchmarks can become fodder for the LLM in the next iteration.

---

> ### Author Response · Authors · 2023-11-23
>
> Thank you for the very useful review. Here are the responses to your concerns.
>
> > I view this work as an additional benchmark to the existing ones that involves a broader set of domains.
>
> Thank you for emphasizing those lines of work! We do cite [1], which is a similar work as ours, developed concurrently.  The main differences are: (1) their quantitative problems are slightly easier than in our benchmark, and require much less deep math knowledge; (2) our math/physics questions are not multiple-choice; (3) their quantitative problems are drawn from a single source which is somewhat more likely to be thoroughly discussed online.
> The updated (Oct 2023) version of [1] also discusses a version of self-evaluation in the “Can GPT-4 find and correct its mistakes” section, and find a negative result using some simple prompts. We introduce rubric-based self-evaluation as a way to make progress on this.
>
> We also cite [3] (the followup to the mentioned [2]), which is indeed quite challenging on a second glance, but is in a different area. We moderated our language (in the paper and in the abstract) to avoid claiming that our benchmark is uniquely challenging for current models.
> > Moreover, it’s not entirely clear how solving the benchmark (especially MCAT or Law questions)  would correlate with reasoning ability.
>
> We agree that the term “reasoning ability” does not have a clear definition in the literature.
> Our estimate is that some of the law problems are quite nontrivial for a non-trained human, and the sources agree. It is possible that the main abilities required to solve it are reading comprehension and pattern matching from memorizing a lot of legal cases, and not “reasoning” as in the math or physics problems.
> > Is a non-parsable answer deemed as an incorrect answer? Or is that question not even considered in the final evaluation?
>
> Thank you for bringing this point up. We do discuss this in the Evaluation section.
>
> For numerical problems, we prompt the model to respond with the final answer, and if the final expression is not parseable using Python’s SymPy library, it is marked as incorrect. (For physics problems, there is the additional step of specifying what units to use for the final answer.)
>
> The process is harder for symbolic answers since they have less structure and are generally harder to parse. We use SymPy to check equivalence between model output and the ground truth up to a permutation of variables. If the two are not equivalent or the model output is unparseable, then the answer is marked incorrect. Manual inspection revealed little discrepancy in the scoring compared to an "ideal" equivalence checker.
>
> > Diagonalization procedure. Given the rapid iterations of LLMs with new and more extensive training data, a diagonalization procedure could be worth investigating,
>
> We do agree that novel methods that maintain utility of all recent benchmarks after publication would be great; however, this was somewhat outside the scope of this project. We try to keep our data behind an API, accessible by requests that are not explicit links, so simple crawlers will not just ingest it; and “smart” LLM-based data collection mechanisms hopefully realize it’s a benchmark. In the current paradigm, it is impossible to prevent overfitting in evaluation if model creators are not careful.
>
>
> [1] Arora, D., & Singh, H. G. (2023). Have LLMs Advanced Enough? A Challenging Problem Solving Benchmark For Large Language Models. arXiv preprint arXiv:2305.15074.
>
> [2] Valmeekam, K., Olmo, A., Sreedharan, S., & Kambhampati, S. (2022). Large Language Models Still Can't Plan (A Benchmark for LLMs on Planning and Reasoning about Change). arXiv preprint arXiv:2206.10498.
>
> [3] Valmeekam, K., Marquez, M., Sreedharan, S., & Kambhampati, S. (2023). On the Planning Abilities of Large Language Models: A Critical Investigation. arXiv preprint arXiv:2305.15771.

---

### Meta-Review · Area_Chair_LkAG · 2023-12-12

**Metareview:**

This paper proposed a new benchmark for evaluating advanced reasoning capabilities in large language models.  Reviewers appreciate that the problems are challenging and cover advanced topics, and the mistakes analysis may help explain why LLMs make an mistake. Main concerns include the evaluation is too complicated, the analysis depends heavily on human evaluation, and the dataset is relatively small. AC agrees that this benchmark needs further polished and extended by addressing the above concerns.

**Justification For Why Not Higher Score:**

The evaluation is too complicated, the analysis depends heavily on human evaluation, and the dataset is relatively small.

**Justification For Why Not Lower Score:**

N/A

---

### Decision · Program_Chairs · 2024-01-16

Reject